



# Grazing mortality as a controlling factor in the uncultured non-cyanobacterial diazotroph (Gamma A) around the Kuroshio region

Takuya Sato[1, a], Tamaha Yamaguchi[2], Kiyotaka Hidaka[2], Sayaka Sogawa[2], Takashi Setou[2], Taketoshi Kodama[2, b], Takuhei Shiozaki[3], and Kazutaka Takahashi[1]

[1] Graduate School of Agricultural and Life Sciences, The University of Tokyo, Tokyo, 113-8657, Japan

[2] Fisheries Resources Institute, Japan Fisheries Research and Education Agency, Kanagawa, 236-8648, Japan

[3] Atmosphere and Ocean Research Institute, The University of Tokyo, Chiba, 277-8564, Japan

[a] Present address: Institute for Chemical Research, Kyoto University, Kyoto, 611-0011, Japan

[b] Present address: Graduate School of Agricultural and Life Sciences, The University of Tokyo, Tokyo, 113-8657, Japan

Correspondence to: Takuya Sato (takusato@scl.kyoto-u.ac.jp)

**Abstract.** Nitrogen-fixing microorganisms (diazotrophs) significantly influence marine productivity by reducing nitrogen gas into bioavailable nitrogen. Recently, non-cyanobacterial diazotrophs (NCDs) have been identified as important contributors to marine nitrogen fixation. Among them, Gamma A is one of the best-studied marine NCDs because of its ubiquitous occurrence; however, the factors controlling its distribution remain unknown. In particular, the importance of microzooplankton grazing as a top-down control has not yet been examined. In this study, we investigated the diazotroph community structure using *nifH* amplicon sequencing and quantified the growth and microzooplankton grazing rate of Gamma A using a combination of dilution experiments and quantitative PCR in well-lit waters at the northern edge of the Kuroshio Current off the southern coast of Japan. In the study region, Gamma A was ubiquitous and dominant in the diazotroph communities, whereas cyanobacterial diazotrophs had lower relative abundances. The microzooplankton grazing rate of Gamma A was significantly higher than that of the whole phytoplankton community and was generally balanced with its growth rate, suggesting efficient transfer of fixed nitrogen by Gamma A to higher trophic levels. Although the in situ growth rates of Gamma A did not show clear responses to nutrient amendments, Gamma A abundance had a significant negative relationship with microzooplankton grazing. This suggests that microzooplankton grazing, rather than nutrient concentration, plays a vital role in constraining Gamma A distribution in the Kuroshio region. Our findings highlight the importance of further in situ quantification of microzooplankton grazing rates to understand the distribution of diazotrophs and its associated nitrogen transfer into the food web.



## 1 Introduction

Biological nitrogen fixation converts inert nitrogen gas into bioavailable nitrogen in the form of ammonia, influencing primary production, and consequently, carbon sequestration through biological pumps in marine ecosystems (Gruber and Galloway, 2008; Karl et al., 1997). Biological nitrogen fixation is performed by specialised prokaryotes called
diazotrophs. These include cyanobacterial and non-cyanobacterial groups, which are commonly detected and quantified using the amplified *nifH* gene, which encodes a subunit of the nitrogenase enzyme (Zehr and Capone, 2021). Cyanobacterial diazotrophs such as *Trichodesmium* and *Crocosphaera* have long been recognised as important marine diazotrophs (Luo et al., 2012; Zehr, 2011; Shao et al., 2023), whereas non-cyanobacterial diazotrophs (NCDs) were also widely detected in the ocean (Bombar et al., 2016; Farnelid et al., 2011; Turk-Kubo et al., 2023) and dominant in *nifH* gene pools in some regions
such as the Arctic Ocean (Blais et al., 2012; Shiozaki et al., 2018b), Indian Ocean (Sato et al., 2022; Shiozaki et al., 2014a), South China Sea (Ding et al., 2021), and South Pacific Ocean (Halm et al., 2012; Moisander et al., 2014). Recently, PCR-free metagenomic and metatranscriptomic studies also revealed the presence and activity of NCDs belonging to diverse phyla such as *Bacteroidota*, *Campylobacterota, Myxococcota, Planctomycetes*, and *Proteobacteria* in the global ocean (Salazar et al., 2019; Delmont et al., 2022; Delmont et al., 2018; Shiozaki et al., 2023).

Among NCDs, Gamma A, also referred to as AO15 (Zehr et al., 1998), UMB (Bird et al., 2005), and γ-24774A11 (Moisander et al., 2008), is the best-studied marine NCD, representing a part of γ-proteobacterial sequences. Compared to cyanobacterial diazotrophs, the physiological and ecological characteristics of Gamma A are poorly understood, mainly due to the lack of isolated cultures and genomic information other than nitrogenase-related genes (Cornejo-Castillo and Zehr, 2021; Turk-Kubo et al., 2023). Still, Gamma A *nifH* is broadly found in open oceans (Cheung et al., 2020; Cornejo-Castillo
and Zehr, 2021; Langlois et al., 2015; Shao and Luo, 2022) and occasionally accounts for the majority of the *nifH* transcript pool (Farnelid et al., 2011; Gradoville et al., 2020; Moisander et al., 2014). Since Gamma A *nifH* usually shows higher abundance and transcripts in well-lit waters in oligotrophic subtropical/tropical regions, it has been speculated to prefer warm and low-nutrient conditions (Langlois et al., 2015; Moisander et al., 2014; Turk-Kubo et al., 2023). On the other hand, a recent meta-analysis using the generalised additive model proposed that Gamma A tends to inhabit cold and highly
productive regions, such as the subarctic oceans (Shao and Luo, 2022). These inconsistent results emphasise our insufficient understanding of Gamma A distribution and the importance of unexplored controlling factors, such as top-down controls like viral lysis or zooplankton grazing (Shao and Luo, 2022).

Recently, model-based studies have proposed that zooplankton grazing is an important factor that controls the global distribution of diazotrophs (Wang and Luo, 2022; Wang et al., 2019). To date, there have been few quantitative
measurements of in situ zooplankton grazing on cyanobacterial diazotrophs. They revealed that the microzooplankton (< 200 μm) grazing rate is generally higher on cyanobacterial diazotrophs than on bulk phytoplankton communities in the North Pacific Subtropical Gyre (Turk-Kubo et al., 2018; Wilson et al., 2017), South China Sea (Deng et al., 2023), and Bering Sea (Cheung et al., 2022) and likely controls the abundances of unicellular cyanobacterial diazotrophs (Wilson et al., 2017),





indicating the potential importance of microzooplankton grazing on NCDs as well. Nevertheless, the microzooplankton
grazing rate on NCDs, including that of Gamma A, has never been quantified.

The Kuroshio Current is a western boundary current of the North Pacific that originates in the North Equatorial
Current, enters the East China Sea, flows across the Tokara Strait, and then runs along the Japanese southern coastal area.
Although the Kuroshio region is generally characterised by warm and oligotrophic conditions (Chen et al., 2009; Kodama et
al., 2014), several nutrient supply processes such as boundary exchange, diapycnal mixing, lateral intrusion, and topographic
flow disturbances create biogeochemical variations along the Kuroshio Current (Hasegawa, 2019; Nagai et al., 2019a).
Previous studies have reported high nitrogen fixation by numerous cyanobacterial diazotrophs (Chen et al., 2009; Chen et al.,
2019; Cheung et al., 2019; Marumo and Nagasawa, 1976; Shiozaki et al., 2014b; Shiozaki et al., 2015b; Shiozaki et al.,
2018a; Wen et al., 2022b). However, these studies were biased toward the upstream region from south Taiwan to southwest
Japan, and downstream areas, such as south mainland Japan, were still undersampled (Fig. S1). In southern Japan, seasonal
contrasts in nutrient concentrations and stable isotope ratios of particle organic nitrogen, a proxy for nitrogen fixation, have
been reported  (Kodama et al., 2014; Kodama et al., 2021), indicating seasonal variation in diazotrophy. Additionally, a
recent study found a dominance of NCDs with a low abundance of cyanobacterial diazotrophs off the southern coast of
Japan in the Autumn (Cheung et al., 2019). These results suggest the importance of NCDs, including Gamma A distinct
regulation of the diazotroph community in the downstream area compared with that in the upstream area. Recently, the rapid
consumption of small phytoplankton by microzooplankton has been suggested as a key factor for the low phytoplankton
biomass in the Kuroshio region near Japan (Kanayama et al., 2020; Kobari et al., 2020). Furthermore, a model-based study
predicted that top-down controls, rather than bottom-up controls, regulate the net growth of diazotrophs in a vast area around
the Kuroshio region (Wang et al., 2019). These previous studies led us to infer that microzooplankton grazing plays an
important role in Gamma A distribution in the Kuroshio region. Therefore, we investigated microzooplankton grazing and
growth rates of Gamma A using a combination of the dilution method and quantitative PCR (qPCR) to understand the
microzooplankton grazing pressure on Gamma A at the northern edge of the Kuroshio region, south of Japan.

## 2 Materials and Methods

### 2.1 Environmental parameters

Our study was conducted during a summer cruise by the R/V *Hakuho-maru* KH-20-09 (10 September to 5 October
2020) and four cruises by R/V *Soyo-maru* (SY2104, Spring [14 April–22, 2021]; SY2109, Summer [3 September–14, 2021];
SY2111, Autumn [17 November–25, 2021]; and SY2201, Winter [12 January–21, 2022]) (Fig. 1, Table 1). The temperature
and salinity profiles were measured using an SBE 911 Plus CTD system (Sea-Bird Electronics, Inc., Washington DC, USA).
Vertical light attenuation was determined using a PRR-600 (Biospherical Instruments Inc., California, USA) or COMPACT-
LTD (JFE Advantech Co., Ltd., Hyogo, Japan). Water samples for DNA, nutrient, and chlorophyll *a* (Chl *a*) analyses were



collected using an acid-cleaned bucket from the surface water and Niskin-X bottles at depths corresponding to 25, 10, and 1 %
of the surface light intensity. Nitrate and phosphate concentrations were determined using a QuAAtro autoanalyser (SEAL
Analytical, Southampton, UK) or an AACSII autoanalyser (Bran+Luebbe GmbH, Norderstedt, Germany). Chl *a* was extracted
using *N, N*-dimethylformamide and measured using a Turner Design 10 AU fluorometer or Trilogy fluorometer (Turner
Designs, California, USA).

Samples (1 L) at the 25 % light depth (13–34 m) for microzooplankton counting were fixed with 2 % Lugol's solution
and stored at 4 °C until analysis. After returning to the laboratory, fixed samples were concentrated using reverse filtration
with a 10-µm mesh. The samples were allowed to settle and condense for at least 24 h. We identified and counted four
taxonomic groups of microplankton communities (naked ciliates, tintinids, dinoflagellates, Radioralia, and copepod nauplii)
as potential grazers, using an inverted microscope (ECLIPSE TS100; Nikon Instruments Inc., NY, USA). The size of the cells
or individuals was measured, biovolume was computed based on the geometric shape, and carbon content was estimated using
conversion equations (Edler, 1979; Mullin, 1969; Putt and Stoecker, 1989; Verity and Lagdon, 1984).

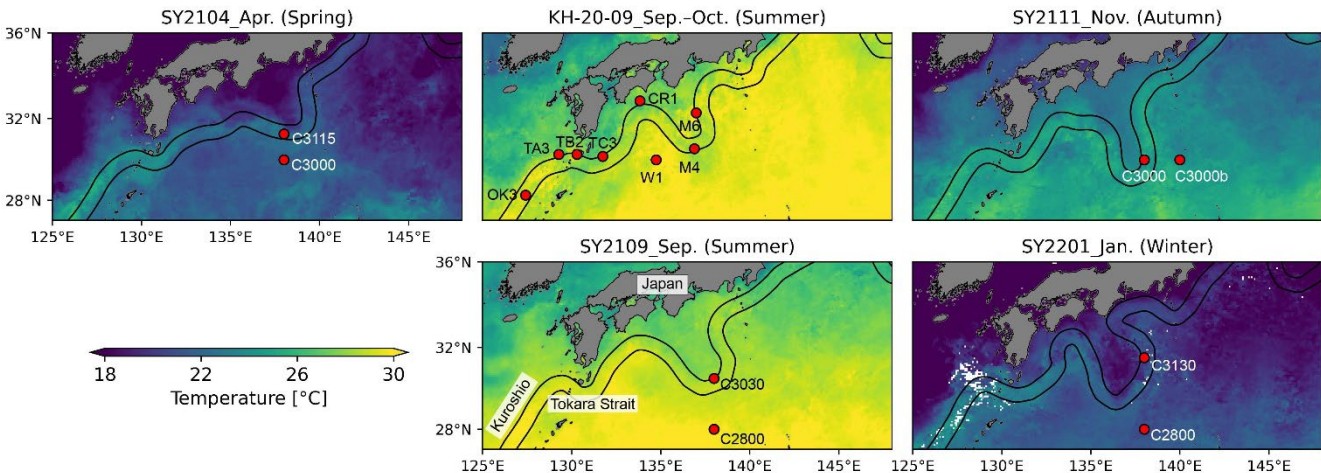

**Figure 1. Sampling stations and the paths of the Kuroshio Current during each cruise. Background contours denote sea surface temperature derived from the satellite (MODIS-Aqua) during each cruise. The black lines represent the paths of the Kuroshio**
**Current during the study periods (http://www1.kaiho.mlit.go.jp/KANKYO/KAIYO/qboc/).**



**Table 1. Summary of environmental variables, Gamma A *nifH* abundance, and microzooplankton (MZ) biomass at the 25 % light depth, in which dilution experiments were conducted, off the southern coast of Japan.**

| Cruise | Station | Date | 25 % light depth (m) | Temperature (°C) | Nitrate ($\mu$mol L$^{-1}$) | Phosphate ($\mu$mol L$^{-1}$) | Chl $a$ ($\mu$g L$^{-1}$) | Gamma A (copies L$^{-1}$) | MZ biomass ($\mu$g C L$^{-1}$) |
|---|---|---|---|---|---|---|---|---|---|
| SY2104 | C3000 | 17 Apr. 2021 | 26 | 21.5 | < 0.04 | < 0.02 | 0.44 | $1.80 \times 10^3$ | 2.92 |
| SY2104 | C3115 | 19 Apr. 2021 | 24 | 22.7 | 0.07 | < 0.02 | 0.64 | $2.60 \times 10^3$ | 4.23 |
| SY2109 | C2800 | 5 Sept. 2021 | 31 | 29.2 | < 0.04 | < 0.02 | 0.06 | $1.91 \times 10^4$ | 0.84 |
| SY2109 | C3030 | 7 Sept. 2021 | 33 | 27.0 | < 0.04 | < 0.02 | 0.17 | $1.31 \times 10^4$ | 1.67 |
| KH-20-9 | OK3 | 13 Sept. 2020 | 16 | 28.7 | < 0.04 | < 0.02 | 0.09 | $1.58 \times 10^3$ | 2.13 |
| KH-20-9 | TA3 | 18 Sept. 2020 | 20 | 29.1 | < 0.04 | < 0.02 | 0.14 | $2.87 \times 10^3$ | 3.33 |
| KH-20-9 | TB2 | 21 Sept. 2020 | 18 | 28.1 | < 0.04 | < 0.02 | 0.37 | $3.00 \times 10^3$ | 3.36 |
| KH-20-9 | TC3 | 26 Sept. 2020 | 20 | 27.9 | < 0.04 | < 0.02 | 0.22 | $3.64 \times 10^3$ | 3.06 |
| KH-20-9 | W1 | 28 Sept. 2020 | 20 | 28.7 | < 0.04 | < 0.02 | 0.14 | $5.41 \times 10^3$ | 0.88 |
| KH-20-9 | CR1 | 29 Sept. 2020 | 20 | 27.8 | < 0.04 | < 0.02 | 0.18 | $3.07 \times 10^3$ | 0.81 |
| KH-20-9 | M4 | 1 Oct. 2020 | 13 | 27.0 | < 0.04 | < 0.02 | 0.14 | $2.44 \times 10^3$ | 1.16 |
| KH-20-9 | M6 | 3 Oct. 2020 | 17 | 27.8 | < 0.04 | < 0.02 | 0.11 | $5.62 \times 10^3$ | 1.20 |
| SY2111 | C3000b | 21 Nov. 2021 | 29 | 24.0 | < 0.04 | < 0.02 | 0.27 | $5.58 \times 10^3$ | 2.12 |
| SY2111 | C3000 | 20 Nov. 2021 | 26 | 23.3 | 0.05 | 0.05 | 0.52 | $1.00 \times 10^3$ | 2.41 |
| SY2201 | C2800 | 18 Jan. 2022 | 34 | 20.5 | < 0.04 | 0.07 | 0.58 | $2.77 \times 10^3$ | 0.49 |
| SY2201 | C3130 | 17 Jan. 2022 | 23 | 21.3 | < 0.04 | < 0.02 | 0.44 | n.d. | 0.49 |

n.d.: Not detected



## 2.2 DNA extraction, *nifH* amplicon sequencing, and qPCR analysis

Samples (2.3 L) for DNA analysis were filtered onto 0.22-μm pore size Sterivex-GP filter units (Millipore, Billerica, Massachusetts, USA) and frozen at –80°C until analysis on land. DNA was extracted using the ChargeSwitch Forensic DNA Purification Kit (Invitrogen, Carlsbad, California, USA).

To reveal the diazotroph community in the Kuroshio region, *nifH* amplicon sequencing (Zehr and Turner, 2001) were performed on all samples from the 25 % light depth. Both the first and second PCRs were performed under the same conditions: 94°C for 2 min, 30 cycles of 94°C for 30 s, 52°C for 1 min, 72°C for 1 min, and finally 72°C for 7 min. Subsequent analyses of diazotroph communities were performed as previously described (Sato et al., 2021). Briefly, PCR products were sequenced using a MiSeq Reagent Kit v3 (600 cycles; Illumina) and sequencing data were processed with QIIME2 program (ver. 2021.4; Bolyen et al., 2019) to produce sequence variants (SVs) using the Deblur plug-in (Amir et al., 2017). For further analysis, the sequences were translated into amino acid sequences, and non-*nifH* and frameshift sequences were removed.

Because *nifH* amplicon sequencing revealed that Gamma A was ubiquitous and most abundant in the sequenced *nifH* pool, we estimated its abundance at all light depths (100, 25, 10, and 1 % light depth) using qPCR assays (Moisander et al., 2008) with the MiniOpticon Real-Time PCR Detection System (Bio-Rad, Hercules, California, USA), as described elsewhere 130 (Shiozaki et al., 2015a). The *nifH* standard was obtained by cloning a known *nifH* sequence. All qPCR reactions were performed in triplicate for each sample. The $r^2$ values for the standard curves ranged between 0.98 and 1.00, and the PCR efficiencies were between 95.6 % and 99.8 %. No signal was detected in the negative group. Based on the extracted volumes, the detection limit was 75 copies $L^{-1}$ seawater.

## 2.3 Estimation of net growth and grazing mortality rates by dilution experiments

Dilution experiments (Landry and Hassett, 1982) with *nifH* qPCR (Cheung et al., 2022; Deng et al., 2023; Turk-Kubo et al., 2018) were conducted to measure the growth and grazing mortality rate of the bulk phytoplankton community and Gamma A at the 25 % light depth. Variations in Chl *a* concentrations and Gamma A *nifH* copies during incubation were used to estimate the growth and grazing mortality rates of the bulk phytoplankton community and Gamma A. It should be noted that the rates of Gamma A were not estimated at St. C3130 in the winter due to an undetectable abundance of Gamma A at the 140 beginning of the incubation.

Seawater from the 25 % light depth was prefiltered with 200-μm mesh to remove mesozooplankton and transferred into acid-washed 23-L polycarbonate carboy. Particle-free seawater, prepared by gravity filtration through an acid-washed 0.2-μm filter capsule (Pall Co., Ltd, New York, USA), was mixed with 200-μm prefiltered seawater in acid-washed 2.3-L polycarbonate bottles at four dilution levels: 25, 50, 75, and 100 % with duplicates. These treatment bottles were enriched with 145 2.0 μmol $L^{-1}$ nitrate (NaNO₃), 0.5 μmol $L^{-1}$ phosphate (KH₂PO₄), and 0.1 μmol $L^{-1}$ iron (FeCl₃) to avoid nutrient limitation





(Landry et al., 2003; Turk-Kubo et al., 2018). Two other bottles filled with 200-µm prefiltered seawater without nutrient enrichment were prepared as controls. The bottles were covered with neutral screens to adjust the light conditions and incubated for 24 h in a water bath with running surface seawater for temperature control. At the beginning and end of the incubation period, all bottles were subsampled for Chl *a* (200 mL) and *nifH* qPCR analysis (2 L). Chl *a* concentration and *nifH* qPCR

analysis for Gamma A performed as described above. Here, we assumed that the number of *nifH* copies per cell did not change during the 24 h incubation period.

Net growth rates in each diluted incubation bottle ($\mu_i$) were estimated as follows:

$$\mu_i = \frac{lnN_t - lnN_0}{t} \qquad\qquad \text{Eq. (1)}$$

where $N_t$ and $N_0$ were the Chl *a* concentrations and Gamma A *nifH* copies at the beginning ($N_0$) and end ($N_t$) of the

incubation period (*t*: d), respectively. $\mu_i$ can be determined with the maximum growth rate ($\mu_{max}$) and microzooplankton grazing rate (*m*) using the following equation (Landry and Hassett, 1982):

$$\mu_i = \mu_{max} - mD_i \qquad\qquad \text{Eq. (2)}$$

where *Di* is the dilution level so that the maximum growth rate ($\mu_{max}$) and grazing mortality rate (*m*) of each prey can be determined with a linear regression of net growth rate ($\mu_i$) against dilution level (*Di*) (Fig. S2). Net growth rates without

nutrient-enrichment ($\mu_0$) were calculated from the nutrient-enriched net growth rates ($\mu_{en}$), corrected for differences in growth rates between amended and unamended treatments of 100 % seawater (Fig. S2). Mortality rates were determined only when significant negative regressions ($p < 0.05$) were obtained; otherwise, they were not calculated. Significant regressions were found in all 16 experiments for the bulk phytoplankton community and in 14 out of 16 experiments for Gamma A.

## 3 Results

### 3.1 Environmental conditions and microzooplankton community

The sea surface temperature showed a clear temporal variation (Fig. 1 and Fig. S3): higher in the summer (September–October 2020 and September 2021), lower in the winter (January 2022), and moderate in the autumn and spring (November 2021). The water temperature at the 25 % light depth, where dilution experiments were conducted, ranged from 20.5 °C to 29.2 °C (Fig. S3, Table 1) and was higher in the summer (27 °C–29.2 °C) and lower in the winter (20.5 °C and 21.3 °C). The

nitrate concentration at the 25 % light depth was generally below the detection limit except at St. C3115 during the spring and St. 3000 during the autumn, showing 0.07 and 0.05 µmol L$^{-1}$, respectively (Table 1). Similarly, phosphate was depleted at most stations, whereas detectable concentrations were observed at St. C3000 in the autumn and St. C2800 in the winter (Table 1). The Chl *a* concentration at the 25 % light depth was lower in the summer and higher in the spring and winter, ranging



between 0.09–0.64 µg L$^{-1}$ (Table 1). The highest and lowest concentrations were observed at St. C3115 in the spring and St.
C2800 in the summer, respectively.

The abundance and biomass of the total microzooplankton community at 25 % light depth ranged 300–3,180 ind. L$^{-1}$
and 0.49–4.23 C µg L$^{-1}$, respectively (Fig. 2; Table 1). Microzooplankton abundance and biomass were higher in the spring
and around the Tokara Strait, such as at St. TA3, TB2, and TC3 in the summer, and lower at St. C2800 in the winter and at the
eastern stations such as St. W1, CR1, and C2800 in the summer. Dinoflagellates and naked ciliates were the two dominant
taxa, accounting for 36 ± 14 % and 34 ± 17 % of total microzooplankton biomass, respectively. Copepod nauplii occasionally
showed relatively high biomass, with a maximum of 57 % at St. M4 in the summer due to their large body size, although their
proportion of abundance was always below 5 % (Fig. 2).

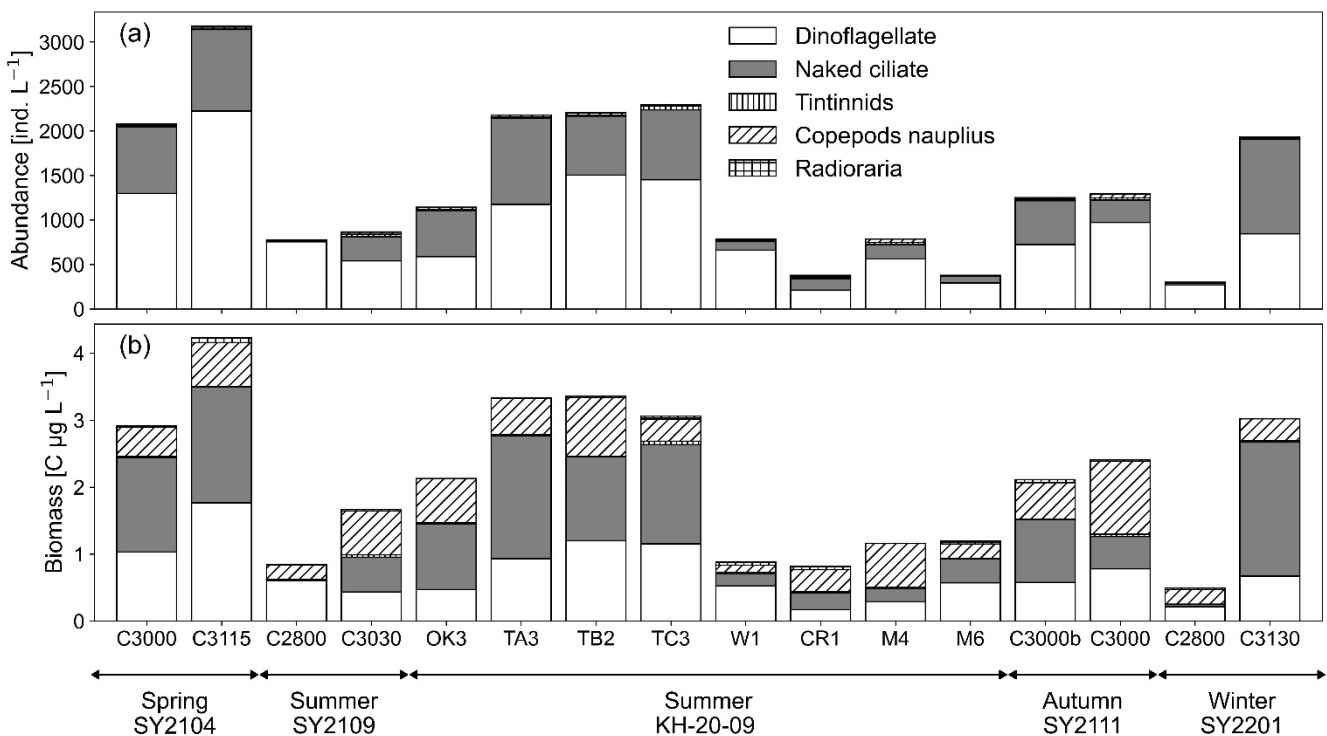

**Figure 2. Microzooplankton (a) abundance and (b) biomass for dinoflagellates, naked ciliates, tintinnids, copepods nauplius, and
Radiolaria at 25 % light depth at each station around the Kuroshio region**



### 3.2 *nifH* gene composition and Gamma A abundance

The *nifH* amplicons from 16 DNA samples at 25 % light depth comprised 70 SVs, of which the 26 most abundant SVs accounted for approximately 97 % of the total sequences at the study stations. Among these major SVs, 21 accounted for 95 % of the total *nifH* sequences, and were affiliated with five well-known diazotrophs: Gamma A (55 %), *Trichodesmium* (18 %), UCYN-A2 (15 %), UCYN-A1 (4.8 %), and UCYN-C (2.9 %) (Fig. S4). The other five SVs included *nifH* Cluster 1B (Cyanobacteria, 2 SVs) and NCDs of *nifH* Clusters 1P (putative δ-proteobacteria, 1 SVs) and 1 J/K (putative α-proteobacteria, 2 SV) (Zehr et al., 2003) (Fig. S4). The most frequently discovered SV, SV001, and 10 other SVs displayed > 97 % similarity at the nucleotide level to Gamma A (Fig. S4). Gamma A was ubiquitous and accounted for 49 ± 22 % of the diazotroph community on average (Fig. 3). At the eastern stations during the summer, such as St. CR1, W1, M4, and M6, Gamma A showed particularly high relative abundances up to 89 % of the total *nifH* sequences (Fig. 3). Cyanobacterial diazotrophs such as *Trichodesmium* and UCYN-A sublineages were occasionally dominant in some samples. UCYN-A1 and UCYN-A2 showed the highest relative abundance, with 57 % of the *nifH* sequences at St. C3000 in the spring and 53 % at St. C2800 in the winter. *Trichodesmium* tended to have higher contributions during the SY2109 cruise and stations west of the Tokara Strait, such as St. OK3 and TB2, during the KH-20-09 cruise in the summer.

The qPCR results also confirmed the extensive occurrence of Gamma A across the study region during the entire season (24 of the 25 stations), ranging from $1.2 \times 10^2$ to $1.9 \times 10^4$ copies $L^{-1}$ (Fig. S3), with a maximum at 25 % light depth at St. C2800 in the summer. Gamma A tended to be abundant in the summer and was undetectable only at St. C3130 in the winter, whose surface temperature (20.6 °C) was lowest among all the study stations. At the 25 % light depth where the dilution experiments were conducted, Gamma A was detected at all stations except St. C3130 and ranged from $1.0 \times 10^3$ to $1.9 \times 10^4$ copies $L^{-1}$ (Table. 1).





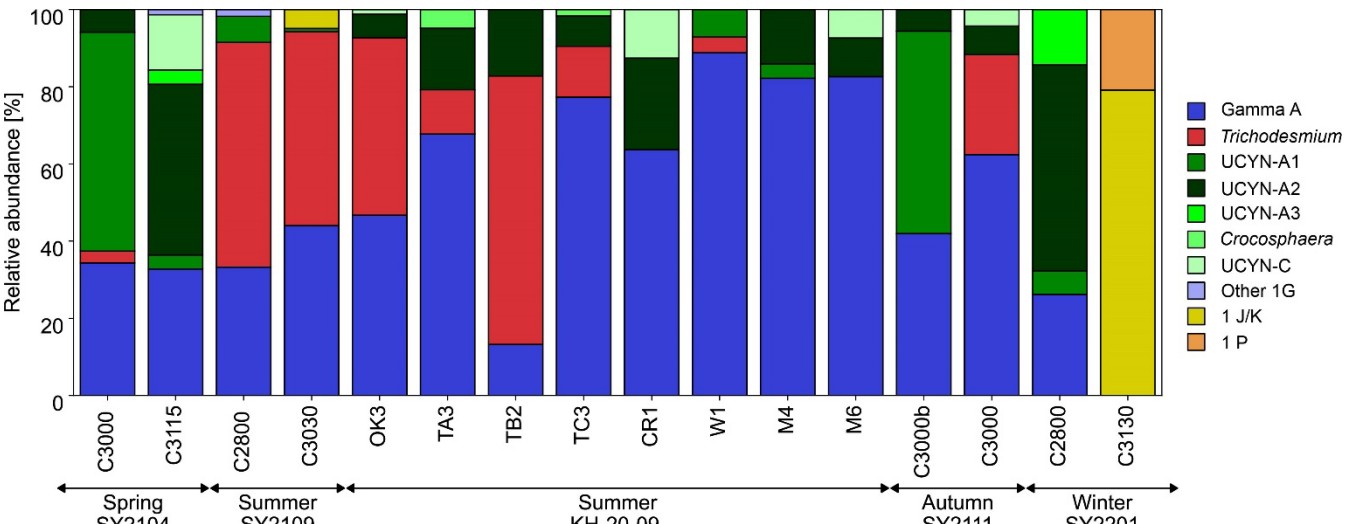

**Figure 3. Diazotroph community composition in the 25 % light depth determined by *nifH* amplicon sequencing in the Kuroshio region.**

### 3.3 Growth and grazing mortality rates of the bulk phytoplankton community and Gamma A

We obtained significant regression lines for all dilution experiments on the bulk phytoplankton community based on changes in Chl *a* ($p < 0.05$; Fig. S2). The grazing mortality rate ($m$: i.e., the slopes of the regression lines in Eq. 2) of the bulk phytoplankton community varied from 0.19–0.82 d$^{-1}$ with an average of $0.45 \pm 0.21$ d$^{-1}$ and was the highest rate at St. OK3 in the summer (Fig. 4 (a), Table S1). The maximum growth rate ($\mu_{max}$: i.e., the intercepts of the regressions corresponding to $D_i$ = 0 in Eq. 2) of the bulk phytoplankton community ranged from 0.60–1.53 d$^{-1}$ with the maximum at the St. C3030 in the summer and the minimum at the St. C3130 in the winter (Fig. 4 (b), Table S1). The net growth rate without nutrient-enrichment ($\mu_0$) and with nutrient-enrichment ($\mu_{en}$) were $-0.10 \pm 0.27$ d$^{-1}$ and $0.41 \pm 0.23$ d$^{-1}$, respectively (Fig. 5 (a), Table S1). $\mu_{en}$ of bulk phytoplankton community was significantly higher than $\mu_0$ (Paired-sampled *t* test; $p < 0.01$, Fig. 5 (a)), indicating that net growth rates of bulk phytoplankton community were generally limited by nutrients in the study area except at St. OK3 in the summer and St. 3130 in the winter where nutrient enrichment did not enhance the net growth rates (Fig. S2).

From 15 dilution experiments based on Gamma A *nifH* abundance, we obtained 14 significant negative regression lines ($p < 0.05$; Fig. S5, Table S1). The grazing mortality rate ($m$) of Gamma A varied between 0.12–1.63 d$^{-1}$ with an average of $0.73 \pm 0.57$ d$^{-1}$, and the highest rates were observed at St. C3115 in the spring (Fig. 4 (a), Table S1). The maximum growth rate ($\mu_{max}$) of Gamma A ranged between 0.37–1.76 d$^{-1}$ with the maximum at the St. SY2104 in the spring and the minimum at the St. C2800 in the winter (Fig. 4 (b), Table S1). $\mu_{max}$ and $m$ had similar trends with a significantly positive correlation with each other ($r = 0.83$, $p < 0.01$, Fig. 4 (c), Table 2). The net growth rate without nutrient-enrichment ($\mu_0$) and with nutrient-




enrichment ($\mu_{en}$) were calculated to be –0.34 to 1.03 d$^{-1}$ (average: 0.40 ± 0.27 d$^{-1}$) and –0.05 to 0.94 d$^{-1}$ (average: 0.39 ± 0.25 d$^{-1}$), respectively (Fig. 5 (b), Table S1). In contrast to the bulk phytoplankton community, there is no significant difference between the $\mu_{en}$ and $\mu_0$ of Gamma A (Paired-sampled $t$ test; $p > 0.05$; Fig. 5 (b)). The $m$ of Gamma A was significantly higher than that of phytoplankton (Paired-samples $t$ test, $p < 0.05$), although $\mu_{max}$ did not show significant difference between Gamma A and the bulk phytoplankton community.


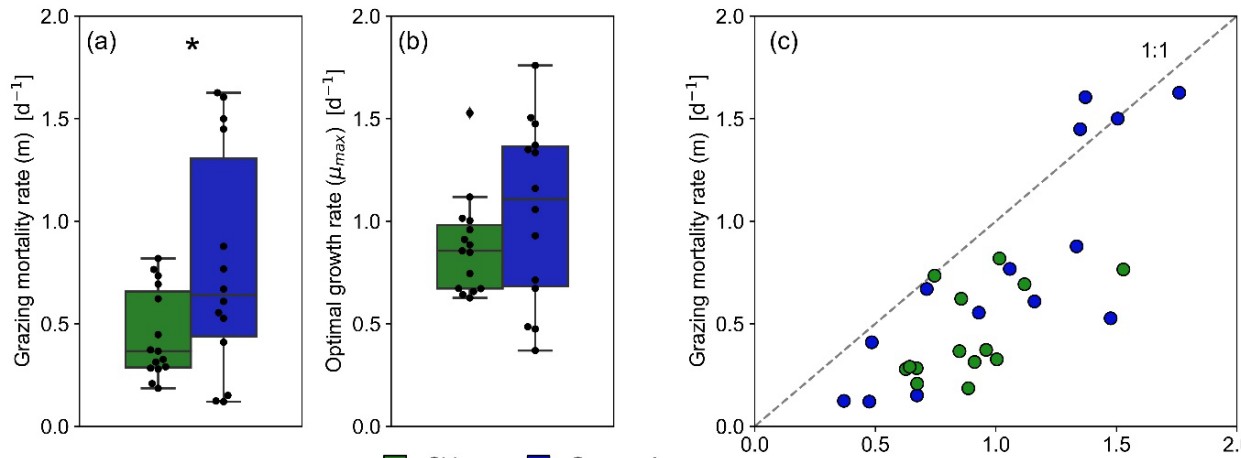

**Fgure 4. Boxplots comparing the (a) microzooplankton grazing rate ($m$) and (b) optimal growth rate ($\mu_{max}$) between Gamma A and the bulk phytoplankton community. The asterisk indicates a significant difference between Gamma A and bulk phytoplankton community (Paired-samples $t$ test; $p < 0.05$). (c) Scatter diagram showing the relationship between $\mu_{max}$ and $m$ of Gamma A and the**
**bulk phytoplankton community.**




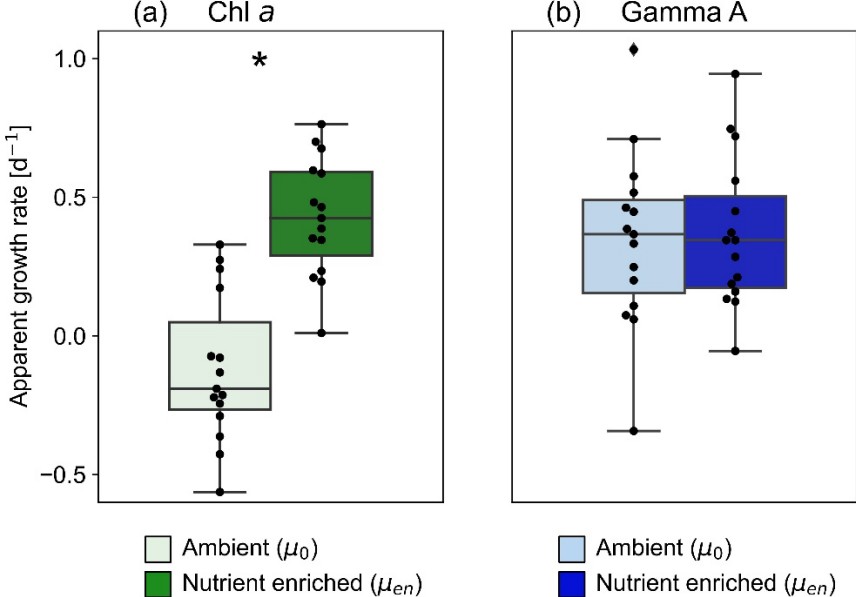

**Figure 5. Boxplots comparing the growth rates under ambient ($\mu_0$) and nutrient-enriched ($\mu_{en}$) conditions of (a) Gamma A and (b) the bulk phytoplankton community. The asterisk indicates a significant difference between $\mu_0$ and $\mu_{en}$ (Paired-samples *t* test; *p* < 0.05).**





### 3.4 Environmental variables explaining distribution of Gamma A

Pearson's correlation analysis with Gamma A abundance and environmental variables showed that the abundance of Gamma A had significant correlations with the temperature ($r = 0.48$, $p < 0.01$; Table 2, Fig. S6), nitrate ($r = –0.37$, $p < 0.01$), and phosphate concentrations ($r = –0.36$, $p < 0.01$), but there was no significant correlation with Chl $a$ ($p > 0.05$). Gamma A abundance has significant negative correlations with the $m$ of Gamma A ($r = –0.57$, $p < 0.05$; Fig. 6 (a), Table 2) and $\mu_{max}$ of Gamma A ($r = –0.52$, $p < 0.05$; Table 2) at the 25 % light depth in which growth and mortality rates were estimated. In addition, the grazing mortality of Gamma A was positively correlated with the biomass of the microzooplankton community (Fig. 6 (b), Table 2).


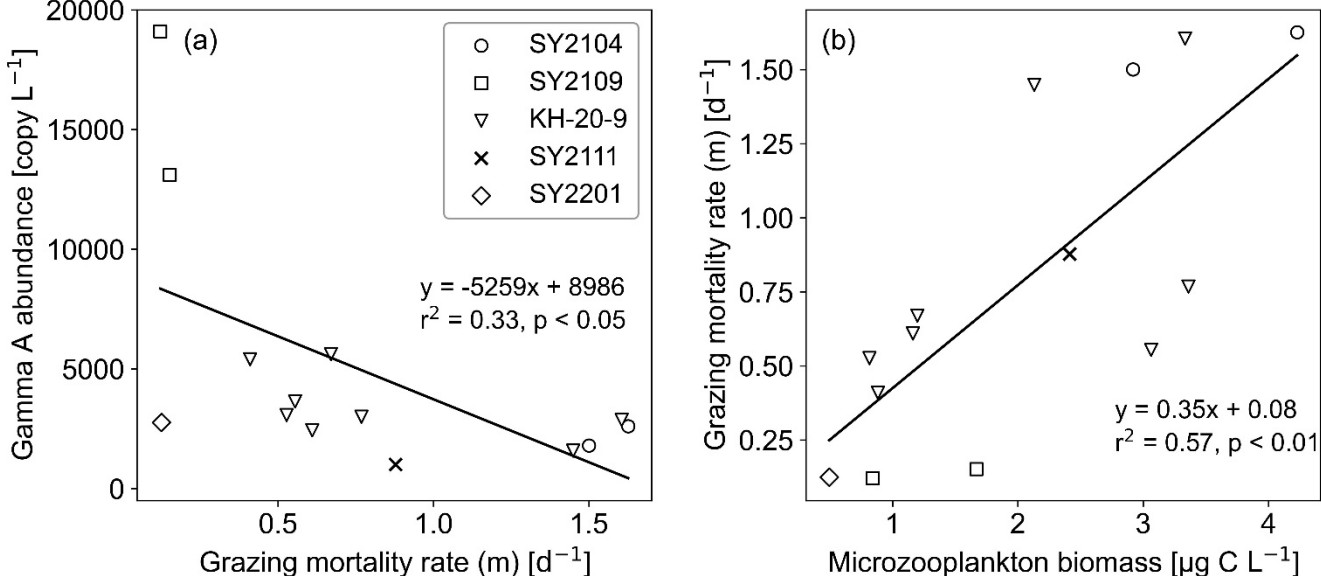

**Figure 6. Relationships between (a) Gamma A and microzooplankton grazing rates and (b) microzooplankton grazing rates and microzooplankton biomass in the 25 % light depth around the Kuroshio region**






**Table 2. Pearson's correlation matrix of Gamma A, each water property, and the parameters derived from the dilution experiments for Gamma A.**

| | Gamma A | Temperature | Nitrate | Phosphate | Chl $a$ | $m^a$ | $\mu_{max}{}^a$ | $\mu_0{}^a$ | $\mu_{en}{}^a$ | MZ Biomass |
|---|---|---|---|---|---|---|---|---|---|---|
| Gamma A | 1 | | | | | | | | | |
| Temperature | 0.48** | 1 | | | | | | | | |
| Nitrate | −0.37** | −0.55** | 1 | | | | | | | |
| Phosphate | −0.36** | −0.54** | 0.99** | 1 | | | | | | |
| Chl $a$ | −0.20 | −0.38** | 0.23* | −0.19 | 1 | | | | | |
| $m^a$ | −0.57* | −0.15 | 0.25 | 0.05 | 0.25 | 1 | | | | |
| $\mu_{max}{}^a$ | −0.52* | −0.19 | 0.38 | 0.18 | 0.30 | 0.83** | 1 | | | |
| $\mu_0{}^a$ | 0.10 | −0.04 | 0.04 | 0.12 | −0.03 | −0.52 | 0.01 | 1 | | |
| $\mu_{en}{}^a$ | 0.01 | 0.06 | 0.30 | 0.38 | 0.03 | −0.39 | 0.15 | 0.78** | 1 | |
| MZ biomass$^a$ | −0.38 | −0.14 | 0.31 | 0.09 | 0.41 | 0.75** | 0.67* | −0.40 | −0.22 | 1 |

**\* $p < 0.05$, \*\* $p < 0.01$**
**$^a$ Since these values were available only at the 25 % light depth, correlation analysis was conducted for samples from the depth.**



## 4 Discussion

### 4.1 Dominance of Gamma A in diazotroph communities in the northern edge of the Kuroshio region

Many previous studies have reported abundant cyanobacterial diazotrophs in more upstream areas than in the present study area (Fig. S1), such as the Philippine Sea (Chen et al., 2019; Wen et al., 2022b), southern Taiwan (Chen et al., 2009;
Cheung et al., 2019; Shiozaki et al., 2014b; Shiozaki et al., 2018a), the East China Sea (Jiang et al., 2018; Jiang et al., 2023; Chang et al., 2000), and southwestern Japan (Cheung et al., 2019; Shiozaki et al., 2015b; Shiozaki et al., 2018a). In contrast, this study demonstrated that Gamma A is ubiquitous and widely dominant in the diazotroph communities around the northern edge of the Kuroshio Current off the southern coast of Japan (Fig. 3). These results are consistent with a recent high-throughput *nifH* sequencing analysis which found a shift in the diazotroph community from cyanobacteria dominated near Taiwan to
NCDs dominated south of Japan (Cheung et al., 2019). They also found an eastward decrease in cyanobacterial diazotroph abundance by two to three orders along the Kuroshio Current. This spatial decline in cyanobacterial diazotrophs may be attributed to the strong nitrate supply along the Kuroshio path, particularly in Tokara Strait (Nagai et al., 2019b; Tsutsumi et al., 2017). The Tokara Strait was reported to have a vertical diffusivity and associated nitrate flux several orders of magnitude higher than those in other Kuroshio areas and other open oceans owing to its steep topographic features (Itoh et al., 2021;
Kaneko et al., 2013; Kaneko et al., 2021; Tsutsumi et al., 2017). This strong nitrate flux alleviates nitrogen limitation on the growth of regional phytoplankton communities (Kobari et al., 2020). Generally, an elevated nitrate supply is considered to suppress cyanobacterial diazotroph abundance because they are outcompeted by non-diazotrophic phytoplankton that utilise resources, such as iron and phosphate, more efficiently (Sato et al., 2022; Ward et al., 2013; Wen et al., 2022b). Consistently, this study found a relative decrease of *Trichodesmium* from the western stations (e.g. St. OK3 and TB) to eastern stations (e.g.
St. CR1 and M4) during a summer cruise KH-20-09 (Fig. 3) and stable abundances of Gamma A in the surface waters between $1.0 \times 10^3$ and $1.9 \times 10^4$ copies $L^{-1}$ throughout the study area, except at St. C3130 in the winter (Table 1). Additionally, nitrate amendments during the incubation experiments enhanced the growth of the bulk phytoplankton community, a potential competitor of cyanobacterial diazotrophs, but did not change Gamma A growth (Fig. 5). This suggests that, in contrast to cyanobacterial diazotrophs, Gamma A remained abundant and became dominant because of its insensitivity to the turbulent
nitrate flux along the Kuroshio Current.

### 4.2 In situ growth characteristics of Gamma A around the Kuroshio region

Little is known about vital rates such as the growth potential of Gamma A mainly due to the lack of established cultures (Turk-Kubo et al., 2023 and reference therein). Previous studies reported that in situ $\mu_{max}$ of unicellular cyanobacterial diazotrophs were higher than or comparable to those of whole phytoplankton communities (Cheung et al., 2022; Deng et al.,
2023; Turk-Kubo et al., 2018). On the other hand, the growth potential of NCDs, including Gamma A, has not been examined despite its potential importance on marine nitrogen fixation. This study revealed, for the first time, that in situ maximum growth rates ($\mu_{max}$) of Gamma A were between 0.37–1.76 $d^{-1}$, which were not significantly different from those of whole phytoplankton



communities (0.60–1.53 d$^{-1}$) (Fig. 4 (b); Paired-samples *t* test, $p > 0.05$). Ecosystem models generally presume that diazotroph growth is slow because of their need to compensate for the high energetic costs of nitrogen fixation (Dutkiewicz et al., 2009;

Ward et al., 2013). The current results, however, indicated that not only cyanobacterial diazotrophs but Gamma A also has growth potential comparable to non-diazotrophic phytoplankton. This highlights the importance of further investigations into the growth potential of diazotrophs, including NCDs, to improve global ecosystem models.

There have been only few studies on in situ nutrient-enriched net growth rates ($\mu_{en}$) of Gamma A (Table S2), and the observed $\mu_{en}$ in the current study (–0.05 to 0.94 d$^{-1}$; Fig. 5 (b)) were in line with the previously reported range in the South

Pacific Ocean (up to 0.52; Moisander et al., 2012) and Southwestern Pacific Ocean (–0.91 to 1.07; Turk-Kubo et al., 2015). Although significant increases in Gamma A abundance by nutrient or dust additions have been sporadically reported from the Southwestern Pacific Ocean (Moisander et al., 2012) and eastern North Atlantic Ocean (Langlois et al., 2012), this study found no significant response in Gamma A growth due to nitrate, phosphate, and iron enrichment (Paired-samples *t* test, $p > 0.05$; Fig. 5). Wen et al. (2022a) reported no significant increase in Gamma A abundance with the amendment of iron and/or

phosphate in the Kuroshio upstream and southeast of Taiwan. This suggests that the growth of Gamma A was not iron- and/or phosphate-limited, and other factors, such as temperature and zooplankton grazing, may be more important for Gamma A distribution in the study region. Further, our study inferred that Gamma A may have a distinct ecological strategy for iron- and/or phosphate-stress from cyanobacterial diazotrophs. In order to confirm their metabolic potential, its genomic contents should be investigated by a metagenomic or isolation approach in the future.

**4.3 Microzooplankton grazing mortality as a controlling factor on Gamma A distribution**

Recently, zooplankton grazing on diazotrophs has received attention as an overlooked controlling factor, because model-based (Wang et al., 2019; Wang and Luo, 2022) and field studies (Cheung et al., 2022; Deng et al., 2023; Turk-Kubo et al., 2018; Wilson et al., 2017) have demonstrated the significance of microzooplankton grazing in the distribution of cyanobacterial diazotrophs. However, knowledge of zooplankton grazing on NCDs is still limited to qualitative gut content

analysis (Scavotto et al., 2015), thus, the importance of grazing pressure on NCDs, including Gamma A remains unknown.

In this study, we quantified, for the first time, the microzooplankton grazing rate of Gamma A using dilution experiments with *nifH* qPCR and found it to be higher than that of the bulk phytoplankton community (Fig. 4(a)). Furthermore, microzooplankton grazing of Gamma A had significant negative relationships with Gamma A abundance and positive relationships with the biomass of microzooplankton communities (Fig. 6, Table 2) which were mainly composed of naked

ciliates and dinoflagellates (Fig. 2). These results suggest that active microzooplankton grazing possibly controls Gamma A distribution. Similarly, high grazing pressure on unicellular cyanobacterial diazotrophs has been reported in the South China Sea (Deng et al., 2023), North Pacific subtropical gyre (Turk-Kubo et al., 2018; Wilson et al., 2017), Bering Sea (Cheung et al., 2022), and in laboratory experiments (Deng et al., 2020). Previous studies (Deng et al., 2020; Deng et al., 2023) have inferred that this active feeding on diazotrophs could be attributed to their nutritious conditions; unicellular cyanobacterial

diazotrophs tend to have lower C:N ratios than non-diazotrophs, which is indicative of high food quality (John and Davidson,



2001). Therefore, Gamma A could also be nutritious prey for grazers, causing higher grazing pressure in the same manner as cyanobacterial diazotrophs. Another possible explanation for the high grazing rate of Gamma A would be its size. Although Gamma A itself may be too small for the prey for microzooplankton, it was found in association with larger size fractions (mainly 3–20 μm), which has raised speculation of a symbiotic- or particle-bound lifestyle (Benavides et al., 2016; Cornejo-Castillo and Zehr, 2021; Harding, 2021). This may make the size of Gamma A suitable for the prey for microzooplankton (10–200 μm in this study), considering the optimal size ratio between prey and naked ciliates (1:8), dinoflagellates (1:1), and copepod nauplii (1:18) (Hansen et al., 1994). This explanation is also supported by reports of greater microzooplankton grazing on nano-phytoplankton (2–11 μm) than other size-fractionated phytoplankton around the Tokara Strait (Kanayama et al., 2020). To validate this explanation, further efforts toward the isolation and in situ visualisation of Gamma A are expected (Harding, 2021). It is also noteworthy that the grazing mortality of Gamma A is nearly balanced by the growth rate (Fig. 4(c); *t* test, $p > 0.05$), whereas the grazing mortality of phytoplankton was significantly lower than its growth rate (*t* test, $p < 0.01$). These results suggest that the nitrogen fixed by Gamma A rapidly transferred to the food web around the Kuroshio Current with a high turnover rate, possibly sustaining regional biological production. Although the cell-specific nitrogen fixation rate of Gamma A has never been reported, future studies combining our results with such parameters will enable us to discuss the quantitative importance of the Gamma A on marine nitrogen cycle.

## 5 Conclusions

This study successfully quantified the grazing mortality rate of Gamma A for the first time, using dilution experiments and *nifH* qPCR analysis. We found that microzooplankton grazing played a significant role in the distribution of Gamma A abundance in the study region, reinforcing the importance of top-down controls on diazotroph distribution. Furthermore, we found that the growth and mortality rates of Gamma A was balanced at higher rates than those of the bulk phytoplankton community. Since Gamma A is a major diazotroph in the study region, it may serve as a starting point for the fixed nitrogen derived from nitrogen fixation into the regional food web. Considering the ubiquitous occurrence of Gamma A subtropical/tropical oligotrophic waters, fixed nitrogen can be introduced into food webs on a global scale. Further in situ quantification of mortality is required to elucidate the importance of grazing pressure on the global distribution of Gamma A. Additionally, other top-down factors, such as viral lysis, should be explored for further understanding of top-down controls on diazotroph distribution and the associated nitrogen fate in marine ecosystems.

## Author Contribution

T. Sato and KT designed the experiments. TSato, TY, KH, SS, and TSetou conducted the onboard experiments and water sampling. TSato, TShiozaki, and TK conducted analyses on land. TSato prepared the manuscript draft with contributions from all the co-authors.



**Competing Interests**

The authors declare that they have no conflict of interest.

**Data Availability Statement**

The recovered sequences were deposited in the DNA Data Bank of the Japan Sequence Read Archive under accession
number    DRA019493.    The    data    used    in    this    study    were    obtained    from    the    UTokyo    Repository
(http://hdl.handle.net/2261/0002009919).

**Acknowledgments**

We would like to thank the captain, crew, and participants of R/V Hakuho-maru and R/V *Soyo-maru* for their
cooperation at sea. We thank H. Saito, H. Morita, and Y. Yoneyama for their kind assistance with water collection and on-
deck incubation. We also thank Kazuya Takahashi for his kind advice on microscopic observations and R. Ebihara for her
helpful comments and assistance with MiSeq sequencing. The authors would like to thank Editage (www.editage.com) for
English language editing. This work was supported by JSPS KAKENHI Grant Numbers 19K06198, 20J22451, 20H03059,
22H03716, 23H02301, 23K26978, and 23KJ1168, the ANRI Fellowship for Young Researchers, and the Fisheries Agency of
Japan through a project titled "Marine Fisheries stock assessment and evaluation for Japanese waters".



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
