# Peer review of "Grazing mortality as a controlling factor in the uncultured non-cyanobacterial diazotroph (Gamma A) around the Kuroshio region"

_EGUsphere, 2024_

## Author Comment (AC1)

Sato et al. conducted a field study to investigate the distribution of non-cyanobacterial diazotrophs (NCDs), especially Gamma A, around the Kuroshio region and used a combination of dilution experiments and quantitative PCR to explore the top-down control of microzooplankton grazing on Gamma A. The topic is interesting and will attract many audiences interested in marine nitrogen fixation. Overall, the manuscript is well-written. However, some issues need to be addressed prior to publication.

We thank Referee#4 for the insightful and constructive comments. In the revised manuscript, we have revised the terminology for growth rates as suggested and added an explanation on the effect of nutrient concentration on Gamma A abundance. Additionally, as recommended by Referee #5, we conducted a generalized linear mixed model (GLMM) analysis instead of a simple correlation analysis to assess the potential factors controlling Gamma A abundance. Details are provided below.

**General comments**

My major concern is about the calculation of the growth rate of Gamma A and the whole phytoplankton community via the dilution technique. The authors mention three kinds of growth rates: the maximum growth rate ($\mu_{max}$), the net growth rate without nutrient enrichment ($\mu_0$), and the net growth rate with nutrient enrichment ($\mu_{en}$). The $\mu_{max}$ is the growth rate of phytoplankton (Gamma A) under nutrient-replete conditions. It is not the *in situ* growth rate because phytoplankton growth could be limited by *in situ* nutrient concentrations. It is reasonable to name it the maximum growth rate, although it is rarely seen in previous dilution papers. The $\mu_0$ and $\mu_{en}$ are the apparent growth rates in the 100% seawater bottles, which involved the effects of microzooplankton grazing. They should equal the instantaneous growth rate ($\mu$) – grazing mortality rate (m). Therefore, to calculate the *in situ* instantaneous growth rate, we can use the net growth rate in the bottles without nutrient enrichment and the grazing rate: $\mu = \mu_0 + m$. Please refer to *Marrec, P., et al. (2021), Seasonal variability in planktonic food web structure and function of the Northeast U.S. Shelf. Limnol Oceanogr, 66: 1440-1458.*https://doi.org/10.1002/lno.11696. To my understanding, comparing $\mu_{max}$ and $\mu$ can reveal the effects of nutrient addition on phytoplankton's growth rate and evaluate whether the *in situ* nutrient condition limits it. By contrast, the $\mu_0$ and $\mu_{en}$ involve grazing loss, and it is hard to identify the effects of nutrients.

We appreciate this comment and agree with Referee #4. We have revised the terminology for growth rates throughout the manuscript. In the revised manuscript, we now use nutrient-amended instantaneous growth rate ($\mu_n$), in situ (without nutrient-amended) instantaneous growth rate ($\mu_0$), and nutrient-amended net growth rate ($k_n$), following Marrec et al. (2021) (L163–172). We also

updated Figure S1 to clarify these growth rates as shown below. Additionally, we compared the nutrient-amended instantaneous growth rate ($\mu_n$) with the in situ instantaneous growth rate ($\mu_0$) to assess the effects of nutrient additions on the growth rates of phytoplankton and Gamma A (Figure 5). Consistent with the original manuscript, we found no significant effect of nutrient amendment.

[Figure]

Part of Figure S1. Schematic illustrating mortality and growth rates ($\mu$n: nutrient-amended instantaneous rate. m: mortality rate by microzooplankton grazing. $\mu$0: instantaneous rate without nutrient-amendment. kn: nutrient-amended net growth rates)

The authors would like to highlight the top-down control of microzooplankton biomass on Gamma A. However, nutrient concentration is one of the important factors. Although nutrient addition did not increase the growth rate of Gamma A in the dilution experiments, it affects the growth rate directly but not the abundance. The abundance of Gamma A is significantly correlated with nutrient concentration, which would indicate the nutrient effects on Gamma A's distribution. Therefore, I suggest discussing this part more comprehensively.

We appreciate the constructive comment. As a result of generalized linear mixed model analyses, nitrate and temperature had negative and positive effects, respectively, on Gamma A distribution across all light depths in the study region. This is consistent with a recent data compilation of global Gamma A abundance (Turk-Kubo et al., 2023). In addition, microzooplankton grazing showed negative effect on Gamma A abundance in the 25% light depth where dilution experiments were conducted. We therefore consider that warm and oligotrophic conditions were prerequisites for Gamma A proliferation, and that under such conditions, active

microzooplankton grazing likely controls Gamma A distribution. We added this discussion at L344–350.

**Specifical comments**

Line 20: microzooplankton grazing mortality rate

Corrected

Line 26: microzooplankton grazing mortality rate of Gamma A

Corrected

Lines 27-28: The nutrient concentration also affects the distribution of Gamma A, as Table 2 shows a significant correlation between them. This sentence seems to negate the effect of nutrients.

We revised the sentence as follows (L27–28).
"This suggests that microzooplankton grazing, as well as nutrient concentration, plays a vital role in constraining Gamma A distribution in the Kuroshio region. "

Line 85: "quantify" should be better than "understand".

Corrected

Line 97: How much seawater was collected for Chl *a* samples? What kind of membrane is used for Chl *a* filtration? Please provide the details.

We added the following sentences about Chl *a* samples collection (Line 98–101).
"For Chl *a* analysis, seawater (250 ml) was filtered on 25-mm diameter Whatman GF/F filters (Whatman, Maidstone, UK). Chl *a* was extracted using *N, N*-dimethylformamide and measured using a Turner Design 10 AU fluorometer or Trilogy fluorometer (Turner Designs, California, USA). "

Line 160: I cannot understand how to calculate the net growth rates without nutrient enrichment. It should be the net growth rate of the 100% seawater without nutrient addition, i.e., the apparent growth rate estimated from these two bottles.

In the revised manuscript, we no longer use the net growth rate without nutrient enrichment. For details on the growth rates used in the revised manuscript, please refer to our response to the General Comment.

Line 227: The average value of the maximum growth rate of Gamma A should be mentioned.

We added the average value for the nutrient-amended instantaneous growth rate (previously referred to as the maximum growth rate in the original manuscript) as follows.

"The nutrient-amended instantaneous growth rate ($\mu_n$) of Gamma A ranged between 0.37–1.76 $d^{-1}$ (average: $1.05 \pm 0.42$ $d^{-1}$) with the maximum at the St. SY2104 in the spring and the minimum at the St. C2800 in the winter (Fig. 4 (b), Table S1). "

Fig. 4: What is the optimal growth rate? It did not occur in the main text.

We apologize for this careless mistake. The term "optimal growth rate" should be replaced with "nutrient-amended instantaneous growth rate ($\mu_n$)". We corrected.

Line 251: the grazing mortality rate of Gamma A

Corrected

Line 251: Add the result of the significance test after this statement.

We added the result of statistical analysis after the sentence.

Line 293: "In situ $\mu_{max}$ "is strange because $\mu_{max}$ is the maximum growth rate with nutrient enrichment of phytoplankton. It is not the growth rate under in situ conditions. Same problem in line 297.

We agreed. We revised as suggested.

Line 300: not only… but also Gamma A has…

Corrected

Lines 312-313: what do " a distinct ecological strategy" mean? Please provide more details.

Thank you for the comment. We added more details like below (L330–333).

"Our study inferred that Gamma A may have a distinct ecological strategy for iron- and/or phosphate-stress from cyanobacterial diazotrophs. For example, as a non-photosynthetic bacterium, Gamma A might have lower iron requirements than cyanobacterial diazotrophs, which require iron for both photosynthesis and nitrogen fixation."

Line 320: change "including" to "especially" and add a comma before "remains".

Corrected.

Line 340: According to Fig. 4c, some points are far from the 1:1 line, indicating lower grazing rates at some stations. Therefore, this statement is imprecise without mentioning these points.

We agree. We revised the sentences as suggested like below (L364–365).

" While Gamma A showed the lower grazing mortality rate than its growth rate at some stations, the grazing mortality of Gamma A is generally balanced by the growth rate (Fig. 4(c); t test, $p > 0.05$)."

Line 347: Delete "successfully"

Corrected.

Line 347: Add "around the Kuroshio region" after "Gamma A".

Corrected

Line 351-352: I cannot understand this sentence. Why fixed nitrogen can be introduced into food webs on a global scale? Please rephrase it.

We apologize for the unclear sentence. Because Gamma A is broadly distributed in subtropical and tropical oligotrophic regions, we consider that nitrogen fixed by Gamma A could be introduced into food webs across wide areas of the ocean. We have rephrased the sentence as follows (L377–379).

"Given the widespread presence of Gamma A in subtropical/tropical oligotrophic waters,

nitrogen fixed by Gamma A may be broadly introduced into food webs, particularly in warm oligotrophic regions."

**Reference**

Marrec, P., McNair, H., Franzè, G., Morison, F., Strock, J. P., and Menden-Deuer, S.: Seasonal variability in planktonic food web structure and function of the Northeast U.S. Shelf, Limnol. Oceanogr., 66, 1440-1458, 10.1002/lno.11696, 2021.

Turk-Kubo, K. A., Gradoville, M. R., Cheung, S., Cornejo-Castillo, F. M., Harding, K. J., Morando, M., Mills, M., and Zehr, J. P.: Non-cyanobacterial diazotrophs: global diversity, distribution, ecophysiology, and activity in marine waters, FEMS Microbiol. Rev., 47, 10.1093/femsre/fuac046, 2023.

---

## Author Comment (AC2)

Sato et al. conducted a field study to investigate the distribution of Gamma A, a non-cyanobacterial diazotroph and the effect of grazing by microzooplankton on Gamma A (and cyanobacterial diazotrophs). As someone with knowledge of the N cycle (including diazotrophy) but not necessarily in microzooplankton and/or grazing, this was a very interesting read and I think it will attract different audiences (i.e., those interested in the N cycle, those interested in NCD and those interested in grazing). The manuscript is well-written, well-prepared and I enjoyed reading it. I only have minor comments.

We deeply acknowledge Referee#5's constructive comments and appreciate the opportunity to respond to them. In the revised manuscript, we have added the result of generalized linear mixed model (GLMM) analyses to assess the potential controlling factors on Gamma A abundance, taking into account the sampling non-independence. Referee#5's comments on our analysis helped us improve our manuscript significantly. We will address the comments one by one as shown below.

**General comments:**

Although it is shortly mentioned upon in the introduction, there are two top-down controls on the distribution of Gamma A: viral infections and zooplankton grazing. In the dilution experiment, the seawater was prefiltered with a 200 μm mesh whereas the particle-free seawater was prepared by filtration through a 0.2 μm filter. Thus, both filtration set-ups would allow for the presence of viral particles. How does the dilution method distinguish between viral lysis or grazing. Does the grazing mortality rate (m) calculated via the dilution method not represent the combined effect of the two top-down processes? Besides that, there also might be an interaction between the grazers and viruses. See e.g.: doi:10.1093/plankt/fbv011. I would like the authors to elaborate a bit more on the potential effect of viruses in their experimental set-up in the discussion.

Generally, the mortality rate calculated from dilution experiments is considered to represent only grazing mortality and does not include viral lysis (Staniewski and Short, 2018; Landry et al., 1998). As Referee #5 noted, both filtration set-ups allow for the presence of viral particles. Therefore, the serial dilution with 0.2 μm-filtered water does not create a proportional gradient of viral particle density (but does create a proportional gradient of grazers, phytoplankton, and Gamma A). This means that the proportional difference in apparent growth rate with serial dilution (slope of regression) should reflect grazing mortality, but not viral lysis. To specifically quantify the viral effect, an additional set-up using 30–100 kDa filtration would be required (known as "modified" dilution experiments), as found in the reference provided by Referee #5 (Pasulka et al., 2015). Potential interactions between grazers and viruses, as discussed in Pasulka

et al. (2015), also become apparent and a problem only through the modified dilution experiments. Therefore, we did not add an explanation on the potential effects of viruses in our set-up. However, since readers may have similar concerns regarding viral impacts in our dilution method, we added the following note to our experimental set-up at L170–172.

"It should be noted that, as viral particles are generally smaller than 0.2 μm and there is no serial gradient of viral density in the diluted bottles, the current calculated mortality rate should theoretically be attributed to microzooplankton grazing mortality (Staniewski and Short, 2018). "

**Statistical analysis**

I do not think a Pearson's correlation analysis is the right statistical method to explain the distribution of Gamma A. For instance, it is mentioned that gamma A has significant correlations with both nitrate and phosphate concentrations. However, there is a very clear correlation between [NO3-] and [PO43-] (see figure S6 and table 2), which could confound the statistical analysis. These measurements are not independent and a mixed-effect model is required for the statistical analysis.

We appreciate referee#5's constructive suggestion. As suggested, we applied generalized linear mixed model (GLMM) analysis to examine the abundance and grazing mortality rate ($m$) of Gamma A incorporating cruises as a random effect. The results of GLMMs were consistent with those of the simple correlation analysis. Temperature had a significant positive effect on Gamma A abundance (coefficient = 1.75, $p$ = 2.3×10$^{-3}$), while nitrate had a significant negative effect (coefficient = –2.17, $p$ = 5.3×10$^{-5}$). Grazing mortality rate ($m$) also had a significant negative relationship with Gamma A abundance (coefficient = –0.39, $p$ = 6.0×10$^{-4}$) at 25 % light depth where dilution experiments were conducted. We revised the method section (L173–187) and result section (L269–278) and updated Table 2 to include the GLMM results.

NifH is a biomarker for potential nitrogen fixation but since the experimental set-up did not include any transcriptomic or proteomic analysis, it remains a potential. I would like the authors to elaborate a bit more about this in the discussion (for instance in chapter 4.2). Does Gamma A have other means to utilize nitrogen-compounds (e.g., ammonium, nitrate, urea) or is nitrogen-fixation the only method to get cellular nitrogen?

We agree. The presence of *nifH* indicates only the potential of nitrogen fixation, and thus transcriptomic or proteomic analysis should be done to confirm the importance of Gamma A as an active diazotroph in the Kuroshio region. We added sentences like below at L305–308.

" Still, it should also be noted that current study only analyzed *nifH* DNA, not RNA or protein, and the dominance in the *nifH* DNA pool does not necessarily mean the most active diazotrophs (Shiozaki et al., 2017). To confirm the importance of Gamma A as an active diazotroph in the study region, future studies should apply transcriptomic or proteomic approaches."

As for other means to utilize nitrogen-compounds, there is no comprehensive information on nitrogen metabolic potential of Gamma A, so its genomic content should be investigated by an omics or isolation approach in the future as mentioned at L332–333.

**Specific comments:**
**Introduction**

l60: in situ in italic

Thank you for the comment, but according to author guidelines 'in situ' should not be italic in the manuscript.

l75: I do not understand this sentence. What stable isotope ratio is a proxy for diazotrophy? Everything containing N has a $\delta15N$ value, but when is this a proxy for nitrogen fixation? Elaborate.

Thank you for the comment. The $\delta15N$ of particulate organic matter (POM) is considered a proxy for nitrogen fixation because it reflects the nitrogen source utilized by organisms. Nitrogen derived from diazotrophs generally shows lower $\delta15N$ than that derived from other autotrophs using nitrate supplied from below the euphotic zone (Minagawa and Wada, 1986; Carpenter et al., 1997). Accordingly, $\delta15N$ and nitrogen fixation have significant correlations in oligotrophic regions like the subtropical waters and Kuroshio (Horii et al., 2018). In the Kuroshio region, Kodama et al. (2021) observed that nitrate depletion and decreased $\delta^{15}N$ values during summer and indicated a contribution of nitrogen fixation to the nitrogen pool in this season. We revised the sentence accordingly (L75–77).

l78-l79: These results suggest the importance of NCDs, including Gamma A, and a distinct….

Corrected.

l83: plays is to strong, change to "might play"

Corrected.

**Results**

Figure 4c: In l229, the significant positive correlation between μmax and m is mentioned (r = 0.83, p < 0.01). Put these statistics in the graph.

We appreciate the comment. We add the statical result in Figure 4c.

Figure S6: Temperature in the second column (not tempereture)

Corrected.

**Reference**

Carpenter, E. J., Harvey, H. R., Fry, B., and Capone, D. G.: Biogeochemical tracers of the marine cyanobacterium *Trichodesmium*, Deep Sea Res., 44, 27-38, 10.1016/s0967-0637(96)00091-x, 1997.

Horii, S., Takahashi, K., Shiozaki, T., Hashihama, F., and Furuya, K.: Stable isotopic evidence for the differential contribution of diazotrophs to the epipelagic grazing food chain in the mid-Pacific Ocean, Global Ecol. Biogeogr., 27, 1467-1480, 10.1111/geb.12823, 2018.

Kodama, T., Nishimoto, A., S, H., Ito, D., Yamaguchi, T., Hidaka, K., Setou, T., and Ono, T.: Spatial and seasonal variations of stable isotope ratios of particulate organic carbon and nitrogen in the surface water of the Kuroshio, J. Geophys. Res: Oceans, 126, e2021JC017175, 10.1029/2021JC017175, 2021.

Landry, M. R., Brown, S. L., Campbell, L., Constantinou, J., and Liu, H.: Spatial patterns in phytoplankton growth and microzooplankton grazing in the Arabian Sea during monsoon forcing, Deep Sea Research Part II: Topical Studies in Oceanography, 45, 2353-2368, 10.1016/s0967-0645(98)00074-5, 1998.

Minagawa, M. and Wada, E.: Nitrogen isotope ratios of red tide organisms in the East China Sea: A characterization of biological nitrogen fixation, Mar. Chem., 19, 245-259, 10.1016/0304-4203(86)90026-5, 1986.

Pasulka, A. L., Samo, T. J., and Landry, M. R.: Grazer and viral impacts on microbial growth and mortality in the southern California Current Ecosystem, J. Plankton Res., 37, 320-336, 10.1093/plankt/fbv011, 2015.

Staniewski, M. A. and Short, S. M.: Methodological review and meta-analysis of dilution assays for estimates of virus- and grazer-mediated phytoplankton mortality, Limnol. Oceanogr. Methods, 16, 649-668, 10.1002/lom3.10273, 2018.